# Long-term spatial patterns in COVID-19 booster vaccine uptake

Anthony J. Wood [1], Anne Marie MacKintosh[2], Martine Stead[2] & Rowland R. Kao [1,3] ✉

## Abstract

**Background** Vaccination is a critical tool for controlling infectious diseases, with its use to protect against COVID-19 being a prime example. Where a disease is highly transmissible, even a small proportion of unprotected individuals can have substantial implications for disease burden and control. As factors such as deprivation and ethnicity have been shown to influence uptake rates, identifying how uptake varies with socio-demographic indicators is critical for reducing hesitancy and issues of access and identifying plausible future uptake patterns.

**Methods** We analyse COVID-19 booster vaccinations in Scotland, subdivided by age, sex, dose and location. Linking to public demographic data, we use Random Forests to fit patterns in first booster uptake, with systematic variation restricted to ∼1km in urban areas. We introduce a method to predict future distributions using our first booster model, assuming existing trends over deprivation will persist. This provides a quantitative estimate of the impact of changing motivations and efforts to increase uptake.

**Results** While age and sex have the greatest impact on the model fit, there is a substantial influence of community deprivation and the proportion of residents belonging to a black or minority ethnicity. Differences between first and second boosters suggest in the longer-term that the impact of deprivation is likely to increase.

**Conclusions** This would further the disproportionate impact of COVID-19 on deprived communities. Our methods are based solely on public demographic data and routinely recorded vaccination data, and would be easily adaptable to other countries and vaccination campaigns where data recording is similar.

## Plain language summary

A key line of defence against many infectious diseases is vaccination, and it is especially important to vaccinate people at the highest risk of becoming seriously unwell if infected. For COVID-19, this includes (amongst others) the elderly and people living in more deprived areas. Looking at Scottish vaccine data and comparing it to data on how deprived different areas in Scotland are, we show how uptake has dropped off in some of these at-risk groups. This is different from the early stages of the pandemic, when there was more motivation to get vaccinated so life could get back to normal. This is concerning because, although we are no longer in a pandemic, COVID-19 continues to circulate in high numbers, and the protection offered by a COVID-19 vaccine is relatively short-lived. The findings of this study suggest that those at-risk groups should be specifically targeted for booster vaccinations moving forward.

Vaccine hesitancy is a critical problem that severely impacts our ability to control important infectious diseases such as measles and seasonal influenza, and has been the subject of much scrutiny during the COVID-19 pandemic. In a voluntary campaign, uptake will depend on individual decision-making, and this is known to be influenced by sociological and demographic factors[1,2]. Quantifying these factors can be challenging, but is invaluable for understanding the context of individual decisions and developing strategies to improve rates of uptake.

Scotland's COVID-19 vaccination programme began in December 2020 and, as of May 2024, had delivered over 14 million doses to a population of 5.4 million, with about 93% of those aged 20+ receiving at least one dose. A primary course for all adults was followed by subsequent rounds of boosters, the first in Autumn/Winter 2021, primarily targeted at ages 50+[3], then rapidly expanded to all adults in response to a wave of the B.1.1.529

Omicron variant of concern (VOC) in November 2021[4]. Regular rounds of vaccination have taken place since, aimed at specific age groups and those otherwise considered vulnerable to severe disease, first in Spring 2022 for all aged 75+[5], then in Autumn 2022 with all aged 50+ eligible[6], and similar rounds every six months thereafter. The pressure on healthcare systems in Scotland from COVID-19 has remained substantial; most recently a wave of infection in Summer 2024 preceded a peak of 485 COVID-19 related hospital admissions in the week ending July 7 2024 (contrasting with an all-time peak of 1069 in w/e April 7 2020)[7]. At the individual level, the presence of additional chronic health conditions (e.g., hypertension, diabetes) increases the risk of severe COVID-19 disease[8]. These conditions are associated with higher age and deprivation, which both manifest as risk factors for severe COVID-19 outcomes at the population level[9–12]. Waning immunity[13,14] and the narrower eligibility criteria for future boosters raise

[1]Roslin Institute, University of Edinburgh, Edinburgh, UK. [2]Institute for Social Marketing and Health, University of Stirling, Stirling, UK. [3]School of Physics and Astronomy, University of Edinburgh, Edinburgh, UK. ✉e-mail: rowland.kao@ed.ac.uk

the question of how best to target these most vulnerable sections of the population.

Belonging to a Black or other minority ethnicity and living in a community with more severe socioeconomic deprivation are consistent risk factors for lower uptake of routine vaccinations (e.g., shingles[15,16] and seasonal influenza in adults[15,17,18], and seasonal influenza[17,19], HPV[20] and MMR[21] in children when considering parents' status). Socioeconomic deprivation is usually presented in these studies in terms of an overall deprivation quintile of the community in which an individual resides, aggregating measures over many different aspects of deprivation (e.g., income, education, health)[22,23]. At the individual level for COVID-19 vaccination, attitudes towards accepting follow-up booster doses over time may differ from those towards the initial course of vaccination, especially as the perceived threat of COVID-19 changes over time. Surveys on hesitancy have highlighted reasons for accepting the first vaccination course that were specific to the context of the pandemic at that time. These include a desire for non-pharmaceutical interventions to end, a feeling of moral duty, and concern of potential requirements of vaccination to travel[24–27]. Hesitancy varied over the early stages of the epidemic[28–30], and in Scotland, uptake has fallen on each successive round of vaccination. It is reasonable to expect uptake to fall further in the future, barring radical changes in pathogen virulence or transmissibility.

Our aims with this paper are twofold. First, we describe for the first and second booster programmes the differences in uptake across demographics and specific markers for socioeconomic deprivation. For first boosters, we use a Random Forest regression model to explain spatial variation in uptake using known risk factors for vaccine hesitancy, refusal and availability, thereby quantifying the importance of these risk factors in a geographically explicit context. Second, we explore a method for using this model for first boosters to predict[31] future spatial variation, should uptake continue to fall and the risk factors for vaccine hesitancy were to persist.

The main findings of this work are that differences in vaccine uptake with socioeconomic deprivation manifest as a spatial clustering of communities with low uptake. Also, owing to the changing eligibility criteria and the context of the pandemic since the first booster programme, the patterns of uptake predicted by our first booster model do not line up with data on second boosters and beyond, indicating changing motivations over time for accepting a COVID-19 vaccine.

## Methods
### Data
Aggregated Scottish vaccination data were provided to the authors by the Electronic Data Research and Innovation Service (eDRIS). eDRIS serves as the point of contact for access to Public Health Scotland's administrative data for research. These data were provided to the authors under a data sharing agreement, approved by the NHS Scotland Public Benefit and Privacy Panel for Health and Social Care (HSC-PBPP). Data were provided at an aggregated level and no individuals are cited, thus there was requirement for informed consent for this work. As these data were used for research only, no additional approval was required to conduct this study.

Individuals are grouped by sex, age range (0–19, 20–29, 30–39, 40–49, 50–59, 60–69, 70+), and residing datazone (DZ; census areas of order 500–1,000 individuals, each with an area as low as 0.15–0.4 km$^2$ in densely populated areas). For each of these groups, the data gives the number of individuals who have received exactly 1 dose, exactly 2 doses, exactly 3 doses, and exactly 4 doses. When the number of individuals is fewer than 5, the exact number is not provided, and we take an estimate (see Supplementary Methods 1).

Population denominators are taken from the census table UV102b, giving small-area populations by age and sex as of 22 March 2022[32]. Data on small-area population breakdown by ethnicity are also obtained from the 2022 census data, table UV201b[33].

Measures of deprivation are taken from the Scottish Index of Multiple Deprivation (SIMD) dataset[23]. These data are publicly available. The SIMD contains measures of different indicators of deprivation at the DZ level (e.g.,

the percentage of residents living in overcrowded housing). The SIMD also ranks DZs by deprivation in each of Access (e.g., broadband speeds, travel time to public services), Income, Employment, Education, Health, Crime and Housing. These ranks are derived from a weighted average of individual deprivation measures (see Supplementary Methods 1 for further details). A DZ with rank 1 is considered to have the highest relative deprivation, and a DZ with rank 6,976 (out of 6,976) the lowest. An overall deprivation rank and decile are also given from a weighted average of all measures.

COVID-19 vaccination in Scotland began with the administration of a first primary dose, and a second primary dose from eight weeks after. Three months from this initial course, adults then become eligible for a first booster dose, commencing in Autumn 2021. A second round of booster vaccination was available to over-75s and those otherwise considered vulnerable to severe COVID-19 disease in Spring 2022. Then, a further round of booster vaccination was available to over 50s and those otherwise vulnerable in Autumn 2022.

These broad-scale trends in the vaccine data used are summarised in Fig. 1.

We distinguish between two characterisations of booster uptake. Overall uptake is the proportion of individuals to have received a booster vaccination. The denominator is the population. Returning uptake is the proportion of individuals who have received at least one dose and have returned for a booster. The denominator is the number of individuals to have received at least one dose. The product of returning booster uptake and overall first dose uptake is then the overall booster uptake. Our model will be fit to returning uptake, but we report in terms of overall uptake where appropriate.

We exclude the 0–19 age bracket, which includes many very young individuals who were not eligible for any vaccine or booster. Finally, a small fraction of individuals with severely weakened immune systems are eligible for additional primary doses, on top of boosters[34]. Due to the structure of the data used here, we define first booster uptake for all individuals as uptake of the third available dose, of any type, and second booster uptake as uptake of the fourth available dose.

### A model for first booster uptake
The 6976 DZs, 6 age ranges, and 2 sexes divide the population into 83,712 subpopulations, each with ~0–100 individuals, and we term these cohorts. We fit a Random Forest regression to cohort-level returning booster uptake. The model is informed by: age range, sex, ethnicity (% population not identifying as white British), and DZ-level deprivation ranks, by Access (e.g. broadband speeds, travel time to public services), Income, Employment, Education, Health, Crime and Housing.

To keep the model simple as well as suitably defined for generating scenarios with lower uptake later, we use DZ-level ranks rather than individual measures of deprivation as explanatory variables (these themselves have a strong degree of correlation, see Supplementary Fig. 1). Details of the individual deprivation ranks that feed into the model are given in Supplementary Methods 1.

**Statistics and reproducibility.** We trained a set of models with different hyperparameters, testing across: number of variables tested per tree split (2, 3, 4, 5), maximum node size (2000, 3000, 4000), and random number seed (7 seeds tested each), and training proportion (70%, 80%). Each model had 1000 trees. We chose the hyperparameters and seed of the model (node size 5000, training proportion 80%, max node size 4000) that explained the most DZ-level variation (R-squared) in the DZs the model was not trained on. Fixing the random number seed ensures the same model is produced in each execution of the code.

### Predicting future distributions of uptake
It is important to account for population heterogeneity when modelling diseases with high variation in susceptibility from person to person. Similarly, it is important that the amount of vaccine-induced protection in demographics especially vulnerable to disease is accurately calibrated. For modelling scenarios where vaccine uptake has fallen, a simple approach

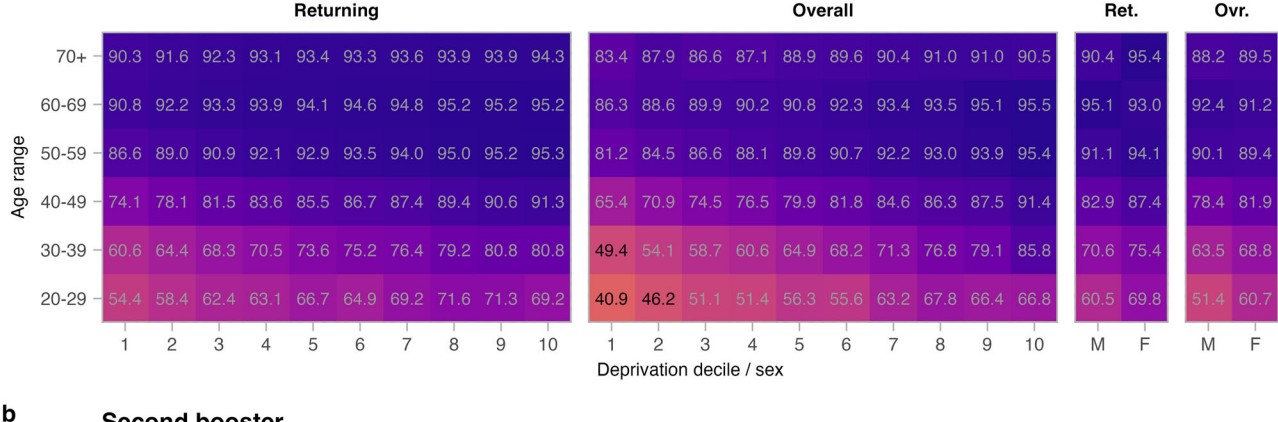

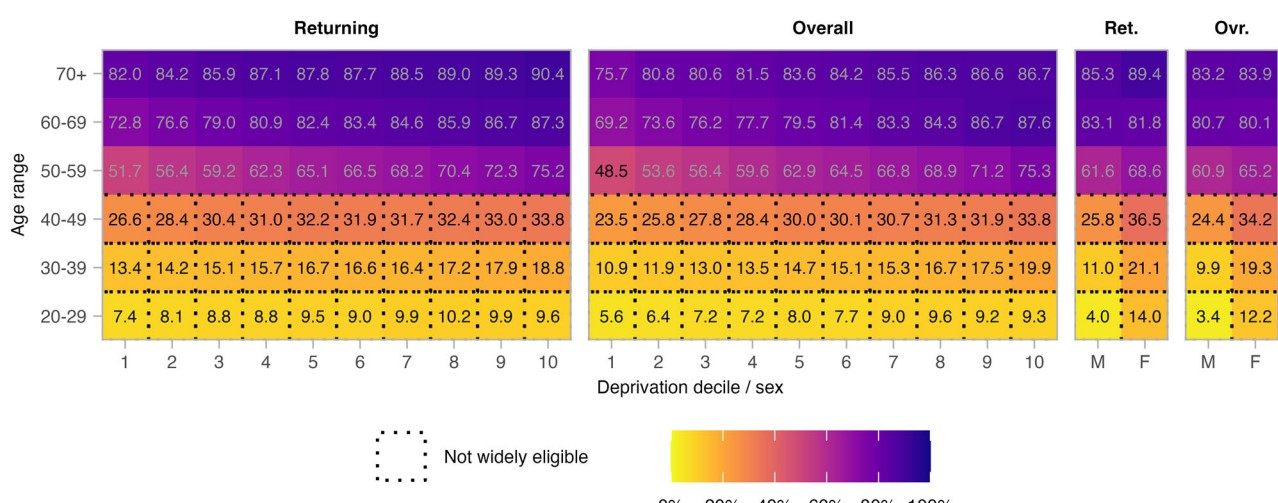

**Fig. 1 | Booster vaccination uptake with respect to age range, sex and datazone (DZ) deprivation decile. a** First boosters. **b** Second boosters. Overall uptake is the proportion of all individuals to have received a booster (the denominator being the population size). Decile 1 contains the most deprived DZs, and Decile 10 the least deprived. Second boosters were not widely available to those aged below 50.

would be to take the uptake from data at some time in the past, and reduce it proportionately across all demographics. However, changes in the observed patterns of drop-off in Fig. 1 suggest a more complex relationship, and in this case, at least, a different approach is required. With limited long-term data, then, we propose a method for redistributing uptake in a non-arbitrary manner, by identifying which fine-scale population groups may be prone to disproportionately larger falls in uptake.

Our regression model fit to first booster uptake takes input data on deprivation and population structure, and reproduces detailed spatial patterns with high accuracy despite not being informed by spatial data explicitly (such as where DZs are located or nearest neighbours). With our model, we are free to feed in data that have been modified in some way, such as data where population structure is unchanged, but the profile of deprivation is different. In doing this, the model will output uptake values that may differ from those using the original data. Such adjustments of the input data form the basis of standard methods of probing machine learning models, such as feature importance (where a variable is randomly shuffled to assess its influence on model performance) and partial dependencies (where one variable is modified to assess its influence on a model outcome)[35].

Along these lines, we propose a method for assessing the risk of different population groups suffering disproportionate falls in uptake in the future. Our hypothesis is that deprivation is the key driver for spatial differences in vaccine uptake, and that population groups whose fitted values are more sensitive to small changes in community deprivation are prone to suffer disproportionately higher falls in uptake. This model is equivalent to proposing that, as the underlying impetus to vaccinate declines, the drop-off in uptake across deprivation cohorts will follow a consistent pattern—i.e., low deprivation cohorts under low vaccination uptake, will follow uptake trajectories similar to high deprivation cohorts under high vaccination uptake, rather than all cohorts following similar uptake trajectories at a given level of vaccine uptake.

We therefore adjust input data in a manner that reduces predicted uptake and assess the resulting distribution. The methodology is similar to a partial dependency analysis but differs in two ways. First, instead of adjusting a single variable, we adjust all deprivation measures (in this case, ranks) in parallel. Second, Random Forest models perform poorly when presented with values exceeding the range to which it was fit (i.e., Random Forest models alone are poor at extrapolation), so we need a means of extending the model prediction for when a counterfactual deprivation rank falls below 1.

The approach is summarised in Fig. 2 with examples in Fig. 3, and detailed in Supplementary Methods 2. For each cohort, we adjust the deprivation ranks of its associated DZ over a range of (positive and negative) values of Δ to give a set of predictions of uptake, to estimate how sensitive the prediction is to changes in deprivation. The parameter Δ here is abstract, without a physical analogue. We then fit to each of these predictions a curve (a sigmoid function) as a function of Δ to these modelled values. By taking

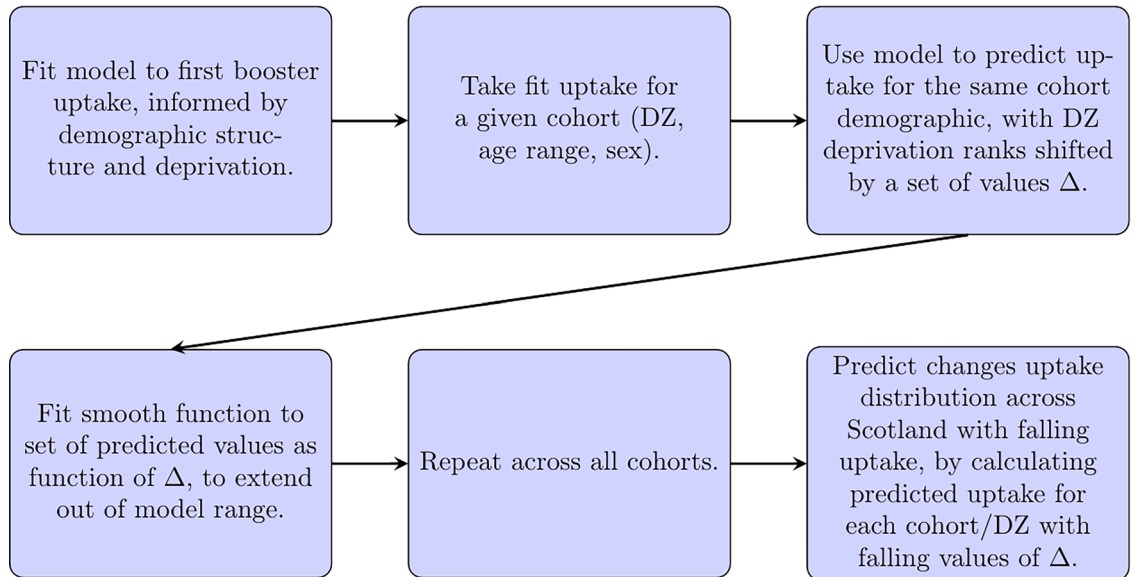

**Fig. 2 | Model fit and prediction methodology.** Flowchart explaining how the fit model to first boosters is used to predict future, lower-uptake distributions in uptake at the cohort level (a group belonging to a specific datazone (DZ), age range, and sex).

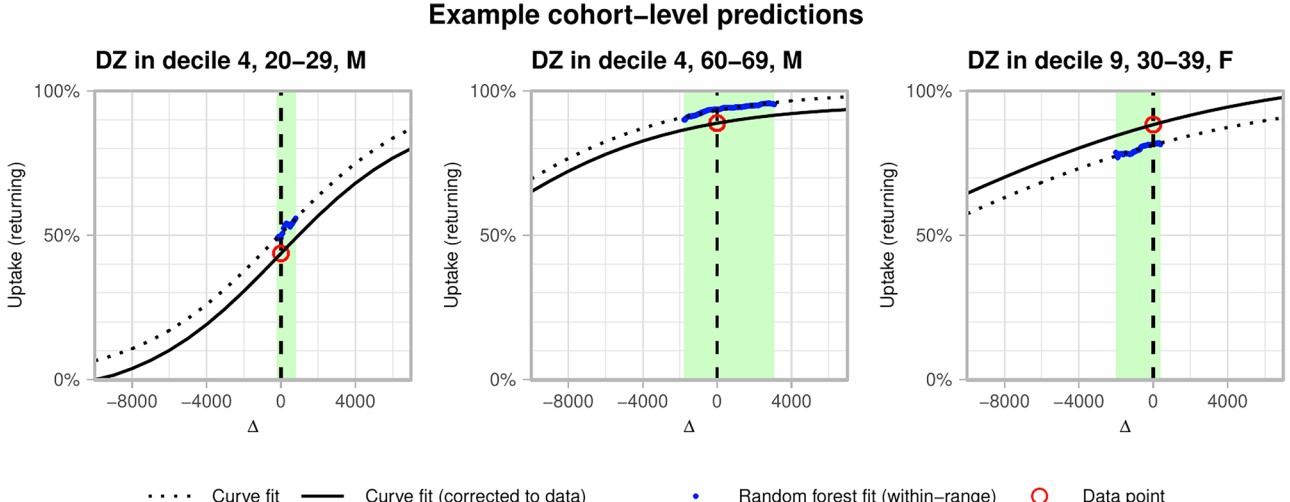

**Fig. 3 | Graphical representation of prediction of uptake with respect to level of deprivation.** For each cohort, the green box bounds the floor and ceiling values of deprivation rank shift Δ. Within this range, the projected uptake (blue points) falls for decreasing Δ (increasing level of deprivation). A sigmoid function (black, dotted) is fit to these fit values, which is then shifted to match the actual returning first booster uptake (red circle) at Δ = 0 (vertical dashed line).

different values of Δ, we then extrapolate distributions of uptake under these counterfactual data. The effect of this functional description is to smoothly extrapolate the deprivation relationships to consider some nominal community with highly severe deprivation (i.e., that would rank below any existing DZ across all deprivation indices). As the value of Δ falls to larger negative values, uptake over all cohorts will fall, but more sharply in those more sensitive to changes in deprivation rank. Conversely, cohorts that are less sensitive to changes will have lower than average falls.

### Reporting summary
Further information on research design is available in the Nature Portfolio Reporting Summary linked to this article.

## Results
### Distributions of first and second booster uptake
First and second booster uptakes are summarised in Fig. 1 with respect to age, sex, and deprivation decile. First booster uptake was 79% in ages 20+

across Scotland overall (returning after a first dose: 85%) and 90% in ages 50+ overall (returning: 93%). Uptake was lower in younger, more deprived subpopulations, as well as in men compared to women. The oldest groups sustained high uptake at all deprivation indices.

For second boosters, focusing on ages 50+ (ages <50 not being widely eligible), uptake fell to 75% (returning: 78%). The skews across each of age, sex and deprivation persist in second booster uptake. We quantify the change in skew with the ratio of returning uptake between (i) men and women, (ii) deprivation deciles 1 and 10, (iii) ages 50–59 and 70+. Comparing the first booster to the second booster, the skew in sex increased from 1.02 to 1.05, in deprivation increased from 1.07 to 1.26, and in age increased from 1.01 to 1.34. Thus, as well as uptake declining from first to second booster, inequalities in each of these metrics were further exacerbated, especially across age and deprivation.

Figure 4a shows relative deprivation in communities over the densely populated central belt of Scotland, and Fig. 4b, c, respectively, show uptake for first and second boosters. The skews in uptake with respect to

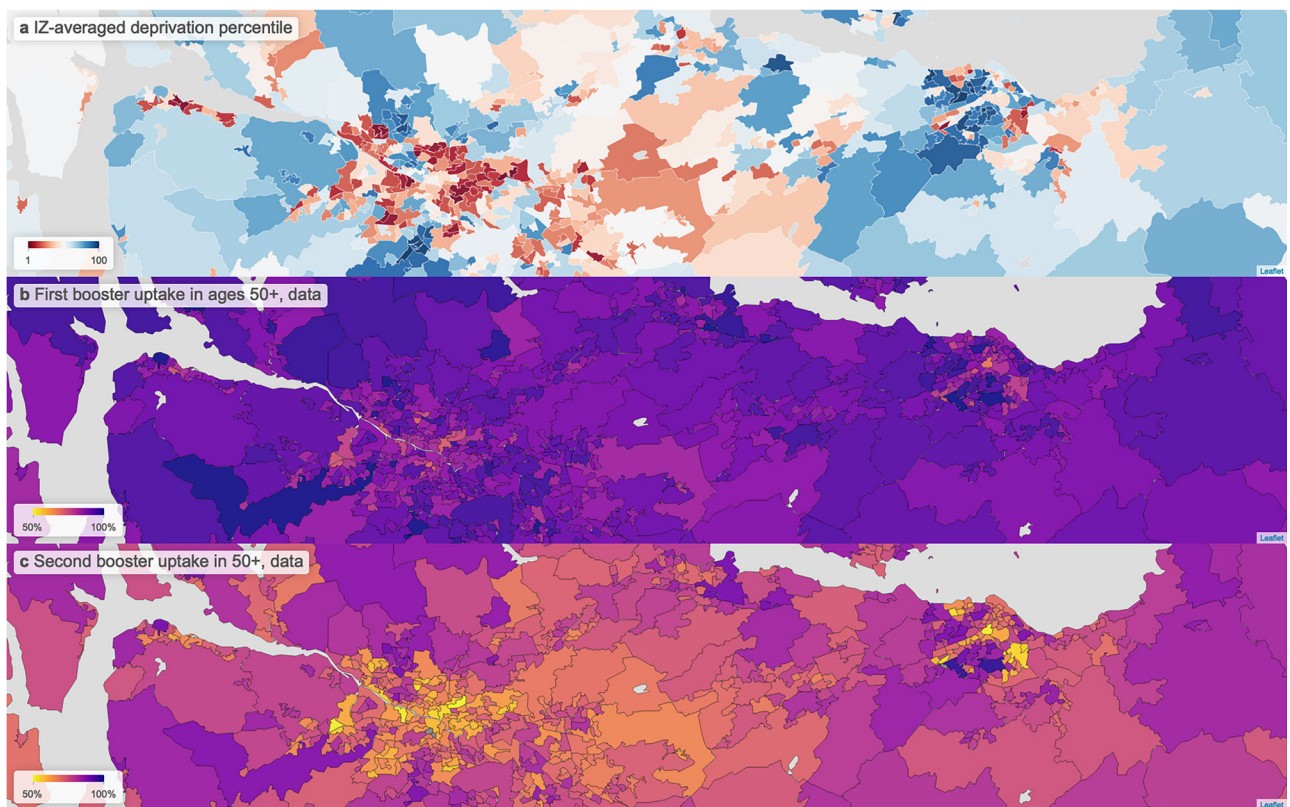

**Fig. 4 | Map views comparing deprivation across the central belt of Scotland, containing the cities of Glasgow (left cluster) and Edinburgh (right cluster), and booster uptake. a** Deprivation. **b** First booster uptake. **c** Second booster uptake. These are presented at intermediate zone (IZ) level; these are geographical areas typically containing 4–6 data zones. Clusters of higher deprivation coincide with clusters of low uptake across both first and second boosters.

deprivation manifest as a spatial clustering of areas with lower uptake, coinciding with communities with higher deprivation.

## Model for first booster uptake

Fitting the random forest model to the first boosters, looking at fit regression values for returning uptake from cohort to cohort, the model explains 83.1% of the variation (fit: 85.6%, test: 74.2%). Aggregating the regression and taking uptake at the coarser overall DZ level, 90.9% of between-DZ variation is explained by the model (fit: 92.7%, test: 83.6%) (Supplementary Fig. 2). There is evidence of clustering in residuals that mostly falls away past distance scales of order 5 km (as measured by the Moran's $I$ statistic[36] in Supplementary Fig. 3), suggesting variation in uptake from local factors beyond the variables used to inform the model.

Variable importance analysis indicates that all variables used serve to improve the model fit (Supplementary Fig. 4), with age the most important explanatory variable, consistent with the empirical trends (Supplementary Fig. 5). To associate a directionality to each of these variables (that is, whether an explanatory variable value increases or decreases the fit value) we also conducted a partial dependence analysis (Supplementary Fig. 6), measuring the influence of different variable values on the average model outcome broken down by age group. Over the deprivation ranks, Education and Housing emerge as having the strongest effects, with a distinct drop-off with increasingly severe deprivation. This is more pronounced in younger cohorts for Education, but for older cohorts for Housing. The partial dependencies also indicate strongly that communities with higher Black and minority ethnic (BAME) populations are, on average, associated with lower fit uptake. Per the census data, fewer than 11% of Scotland's DZs have a BAME population of over 25%. The data do not specify individual-level ethnicity so we can not infer a direct relation here; however, the relation found is consistent with empirical studies finding lower relative uptake in BAME communities[37–40]. Nonetheless, the inclusion of ethnicity (measured here as the percentage of a cohort belonging to a Black or minority ethnicity)

improves the model performance, with a high node purity (Supplementary Fig. 4) indicative of a stronger influence on cohorts where the 20+ BAME population is higher than the DZ mean of 11%.

## Model prediction vs. second boosters in 75+, August 2022

To predict how the distribution of uptake may change under the assumption that existing trends with deprivation persist, we first compare a model prediction to the distribution of second boosters, first in ages 75+, from a snapshot dated August 2022. This snapshot is from a version of the vaccination data from an earlier, since-expired data sharing agreement (see Data Availability for details). We then assess the model prediction on the most recent data. By comparing performance on two sets of data separated in time, we can assess how the model predicts variation in uptake at a time relatively close to the first booster rollout that the model was fit to, and to patterns in uptake at a later time as the acute phase of the pandemic had subsided.

Returning uptake for 75+ at this point as of August 2022, was 82%. Figure 5 shows that the skew with respect to deprivation is well captured, with a predicted ratio of uptake between the highest and lowest deprivation deciles of 1.16, compared to an actual ratio of 1.18.

## Model prediction vs second boosters in 50+, May 2024

With the more recent data from May 2024, returning second booster uptake in ages 50+ is 75%, as compared to 90% for first boosters. An uptake of 75% corresponds to a value of $\Delta \approx -7290$ (an 50+ uptake of 75% is predicted by our model if all DZs were $\approx 7290$ ranks lower). Figure 5 shows our modelled distribution with respect to age, sex and deprivation as compared to actual uptake. The model predicts increased skews with respect to sex (ratio 1.04), age (1.16) and deprivation (1.39), relative to first booster uptake. However, the predicted skew with respect to age is an underprediction (whereas in reality uptake fell especially sharply in 50–59), and overpredicts the skew with deprivation (whereas in reality the trend was less pronounced).

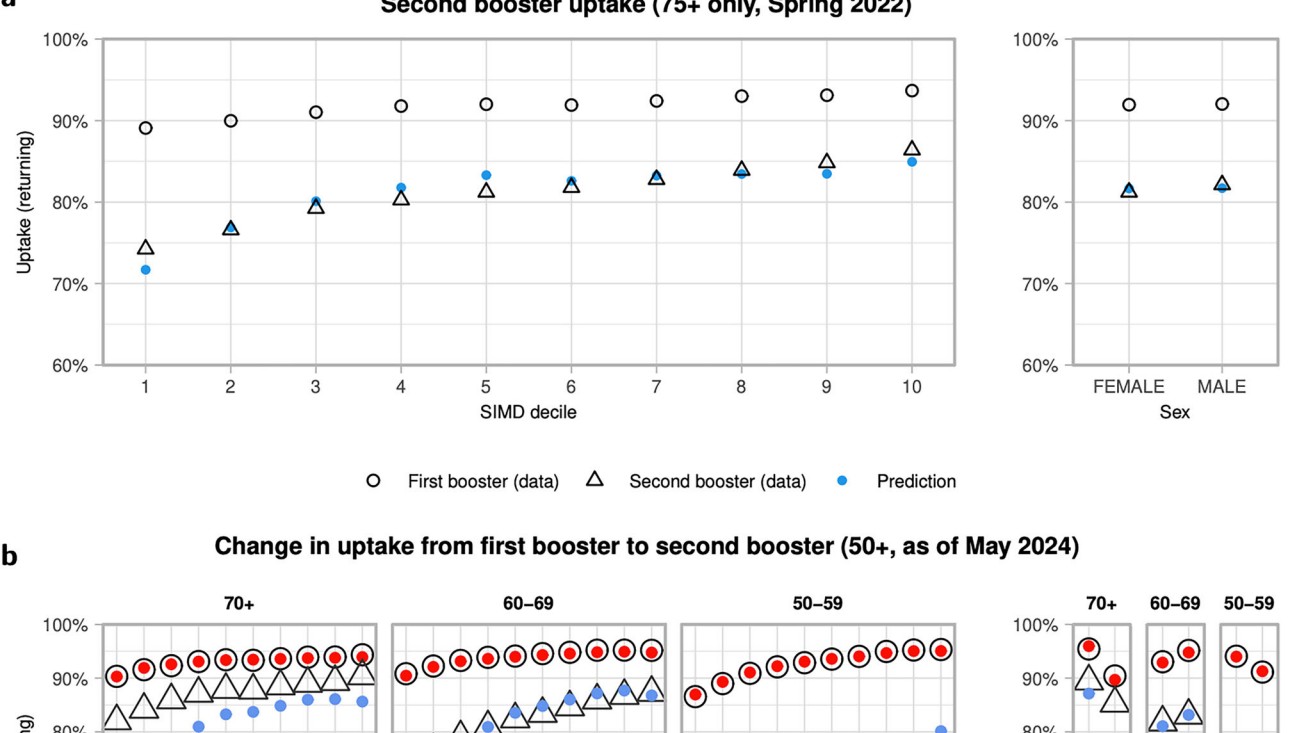

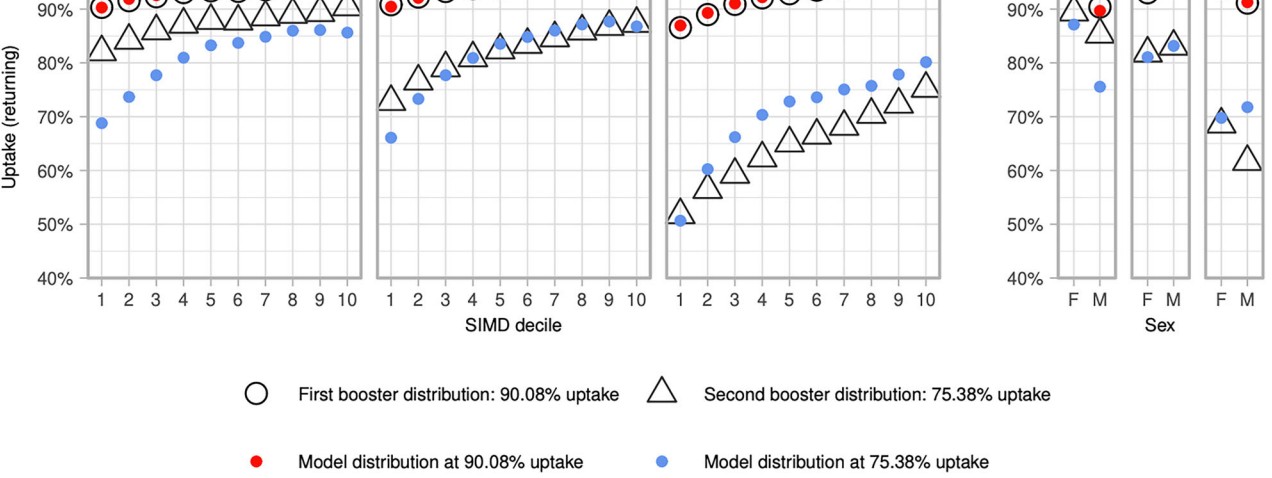

**Fig. 5 | Comparison of returning booster uptake from two different snapshots in time. a** Returning second booster uptake amongst individuals aged 75+, as of August 2022. Filled points are predictions from a model trained on first booster data, when matching the lower uptake in second boosters. **b** Returning uptake in the second booster rollout as of May 2024 across ages 50–59, 60–69, 70+, broken down by sex and deprivation decile, and the corresponding predicted model distribution when matching the second booster uptake of 75%.

## Discussion

We have used high-resolution data to describe patterns in COVID-19 booster vaccination uptake across communities in Scotland. First booster data reveal falls in uptake in younger, more deprived populations relative to the initial vaccination course. This is consistent with observed increased vaccine refusal in younger groups living in more deprived communities[37,41,42]. These inequalities in uptake then manifest as larger clusters of low uptake coinciding with communities with higher general deprivation. The second boosters are the most recent we have data on, and are the best representation of typical uptake to be expected outside of a pandemic scenario. These data show that, as well as a general fall in uptake, skewness seen in first boosters is exacerbated further, mainly by age.

To better understand fine-scale differences in first booster uptake, we have fit a Random Forest regression model, informed by local population structure and deprivation. This explains substantial spatial variation in returning booster uptake, and accurately captures differences by age, deprivation and sex. A community's booster uptake can therefore be estimated with high accuracy solely from its population structure and relative

level of deprivation, without information on where that community is physically located in Scotland, or who its neighbours are.

We then explored a method for predicting future distributions of uptake, using our first booster model. We use prediction here in the sense of prediction forecasting, i.e., to provide estimates of what might actually occur in the future, under the assumption that the underlying conditions of the prediction do not change[31]. We created uptake distributions using counterfactual data, where population structure was unchanged but each cohort had more severe deprivation than it does in reality, fitting a relation between estimated uptake and changes in deprivation. Population groups whose model prediction was more sensitive to small changes in deprivation are interpreted as being at increased risk of falls in uptake.

When set to predict the distribution for the single age group that was eligible for a second booster in Spring 2022, when conditions for vaccination were most similar to the first booster period (high awareness with an intensive campaign to boost uptake), the model successfully captures the increased skew with respect to deprivation. However, when predicting the distribution of second boosters as of May 2024 across multiple age groups,

the prediction is poor compared to the data, and over-estimates the skew with respect to deprivation, and under-estimates that with age, suggesting the underlying risk factors for vaccine refusal were not the same between first and second boosters. This is consistent with the changed context of the pandemic by the time second boosters were administered in the Autumn 2022; for first boosters, eligibility was rapidly expanded from ages 50+ to all adults in response to an outbreak of the B.1.1.529 Omicron variant of concern which saw the re-imposition of some non-pharmaceutical interventions especially in mass gatherings and hospitality[43,44]. However, throughout 2022, these were eased, including dropping the requirement for a vaccine passport in some hospitality settings[45], and global travel restrictions were gradually relaxed. The sharper drop-off with age than predicted by our model fit to first boosters is consistent with a broader shift in COVID-19 strategy towards protecting those deemed most vulnerable[46].

To probe the sensitivity of different cohorts to changes in deprivation using data within the range the model was trained on, we then proposed a method for extending to counterfactual data that effectively exceeds this range, which has provided useful insight. Such an extrapolation as implemented here is necessary as Random Forests perform very poorly with data beyond its training range. Nonetheless, this demonstrates how such a statistical model can be exploited in a way to explore counterfactual scenarios, and indicate when underlying risk factors may have changed.

The fall from first to second booster uptake, combined with an increased inequality with deprivation, highlights a potential twofold risk in the longer term. A fall in uptake obviously reduces the amount of vaccine-induced protection against severe COVID-19 disease in the population. Compounding this, though, with a growing skew in uptake, any shrinking pool of protection may become increasingly associated with those living in less deprived neighbourhoods, already at lower prior risk of developing severe disease. This more substantial fall in uptake from a lower baseline would see clusters of communities that are most vulnerable disproportionately more exposed, and at higher risk of infection spread, hospitalisation and mortality.

Finally, for the purposes of fine-scale infectious disease modelling, the profile of vaccine-induced immunity is one of the few inputs that can be done accurately at the individual level with existing data. With early COVID-19 the exception, epidemiological data are typically sparser, and often representative of only a subset of infections that are severe (hospital/intensive care admissions, mortalities). We have shown here how such data can be used to inform how uptake may be modelled in the longer term, under a set of assumptions. However, despite being routinely collected for many infectious diseases, these vaccination data are not often made available at such a fine spatial resolution to researchers. Our access in this instance has been exceptional, owing to the need for rapid, policy-relevant analysis during the COVID-19 pandemic[47]. Our insights from these data have been entirely done using public data, using methods widely applicable to diseases beyond COVID-19, and to other indicators for vaccine hesitancy and accessibility beyond deprivation. With the wealth of public demographic data available, our results provide a strong argument for the value of access to such vaccine uptake data for COVID-19, as well as other infectious diseases where there is a strong public health interest.

## Data availability
Population denominators can be found in the 2022 Scottish census table UV102b[32], giving small-area populations by age and sex as of 22 March 2022. Data on small-area population breakdown by ethnicity can be found in table UV201b[33]. Both tables are available from https://www.scotlandscensus.gov.uk. Datazone level deprivation measures and ranks are available from https://simd.scot. The vaccination data utilised in this work are not publicly available. They are provided to the authors for academic research by Public Health Scotland's electronic Data Research and Innovation Service, under a data sharing agreement (Spatial and Network Analysis of SARS-CoV-2 Sequences to Inform COVID-19 Control in Scotland), and can be contacted at phs.edris@phs.scot. Two versions of the vaccination data have been used in this analysis; the first specifying at a pseudonymised individual level, each individual vaccination event up to 18

August 2022, which produced Fig. 5. The authors no longer have access to this version after expiry of the DSA. The second version used for all other analyses, as detailed in Data, is a version aggregated into age/sex/DZ population cohorts, giving time-aggregated doses as of May 2024. As of September 2024, this DSA remains active. All other data utilised in this work are publicly available, drawing primarily from Scottish census data.

## Code availability
Model code is available at https://doi.org/10.5281/zenodo.15342303[48] which runs on a synthetic vaccination dataset. There are no restrictions to access of the code with synthetic data.

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

## Acknowledgements

We thank eDRIS for the provision of Scottish COVID-19 vaccination data. We also thank the reviewers for their helpful feedback, which has led to substantial improvement of the manuscript. This work has been funded by the ESRC grant ES/W001489/1: Real-time monitoring and predictive modelling of the impact of human behaviour and vaccine characteristics on COVID-19 vaccination in Scotland.

## Author contributions

R.R.K. conceived the project. A.J.W. wrote the model code, performed the model analysis, and, with R.R.K., wrote the manuscript. A.M.M. and M.S. motivated conceptual work on evolving motives for vaccination and risk factors for declining uptake. R.R.K. and A.J.W. conceived the method for generating lower-uptake scenarios. All authors commented on and approved the manuscript.

## Competing interests

The authors declare no competing interests.
