## [Transparent Peer Review file · Communications Medicine]

Long-term spatial patterns in COVID-19 booster vaccine uptake

Corresponding Author: Professor Rowland Kao

Version 0:

Reviewer comments:

Reviewer #1

(Remarks to the Author)

This paper presents an analysis of a very finely scaled data set for COVID-19 vaccination in Scotland. Specifically, it considers first and second booster uptake stratified by fine spatial scale, age, gender and socio-economic factors. There are three main claims:

- (i) 1st Booster uptake predicts 2nd booster uptake (though at lower general level)
- (ii) Location factors are important for explaining variability in uptake, specifically measures of deprivation and also BAME proportions
- (iii) If uptake decreases, socio-economic factors become relatively more important

The first result is unsurprising but reassuring. The second result is also unsurprising, but very important and there are few studies with the depth of data to explore this well. The third result implies that the disproportional effects of COVID-19 would be exacerbated, which is clearly important.

The authors are careful not to claim application beyond the specific circumstance studied, but in practice what is learnt from this data will potentially be informative for much wider application.

I set out some queries and concerns below. Of all the points below, point 3 is the one that I have the most concerns about, though some aspects of it certainly could be my misunderstanding.

Despite these concerns, this is potentially an important paper as it gives an analysis of an exquisite dataset, and any insights (even with many caveats) about vaccine uptake variation across the population remains among the important things that we should learn from COVID-19, with potential consequences for vaccine control for further COVID-19 variants and other diseases in future. The ability to explore in superb depth here is unique as far as I have seen so far (see for example the work on the residuals in fig 6 by space, as well as the detailed consideration of different deprivation ranks). The approach taken overall could indeed be of use in adapting to other datasets, as the authors mention in the abstract.

1) The population denominators:

The caution that the 2011 census data is used is noted, and is surely a reasonable approach. Is it possible though to give any early estimates about how far these differ from reality in 2021?

Looking at mid-year estimates on National Records of Scotland (e.g. via <https://scotland.shinyapps.io/nrs-population-estimates/>) it looks like the older population age groups are growing, consistent with an aging population.

The figures show a clear anomaly for the age band 70-74 (e.g. Fig 1 second panel), suggesting that uptake appears low as the denominator is erroneously large. But why this one band is so different, and why the wrong way around from the general picture above of an aging population (so the 2011 denominators should be too small at this age group)? Is there any other reason why that age band is such an obvious anomaly?

2) Interpretation of results for BAME:

Last paragraph of section 3 highlights variability in Scotland of BAME proportions across DZs, and that previous studies have highlighted lower vaccine uptake in BAME communities. The results given verbally in that paragraph confirm that for the fit here, proportion BAME in a DZ is selected by the model. But, the strength of the effect, or even the direction of the effect is not given.

While I know this can't just be lifted as a single number out of a random forest approach, every other factor involved is illustrated somewhere – is it possible to give something, e.g. uptake by proportion BAME (possibly binned), given as data and model?

As things stand, the paper justifies the claim that proportion BAME is a factor that influences uptake, but nothing at all on the direction of the effect. Seems important, given that the overarching motivation of identifying how vaccine hesitancy varies across the population.

3) This is my main (and only) major concern: how the model is used to extrapolate to generate a counterfactual.

Here is my understanding: as this is a statistical random forest model, there is no direct way to tweak a mechanistic parameter to simulate a general broad reduction in vaccine uptake. So, the authors have taken an approach of supposing that everyone is moved to a lower deprivation rank (with a fix to stop this going less than 1), and this is used to create a lower uptake.

My concerns are two-fold. (i) Firstly that this seems odd decision on approach, on the face of it, and not clearly justified. (ii) Secondly, I wonder if there's a danger of circularity or bias on the result that deprivation becomes more important if uptake drops.

Expanding on these: (i) this whole approach to simulating a reduction in uptake via turning up deprivation doesn't seem to have been justified. It does achieve the desired outcome of a lower uptake output from model. But why via deprivation? Alternative (equally unjustified) would be to do the same with age (pretend everyone is X years younger, with a fix to keep them over age 20). (One of many problems with that is that it would be extrapolating from semi-fit for the 75+ 2nd boosters to other age groups.) But also why not an even simpler approach: suppose a person in a given cohort has probability P of getting first booster, then if we reduce a pressure/hazard of getting vaccinated, then we can scale new P as $1-(1-P)^x$ for some scaling parameter x. This does bake in a model of how vaccine hesitancy will scale between different people. I'm not sure I'm suggesting exactly this model, but how is this less justified than assuming hesitancy scales like everyone shifts deprivation factor in the given way?

(ii) One of the key points of this paper is that as uptake decreases, socio-economic factors become more important, and vaccine uptake even more disproportionate between groups. However, given the way reduced uptake is simulated is to project from tweaking deprivation rank, this result appears compromised by the approach being so entangled with the claimed effect. In particular, the handling of the most deprived groups requires a fix, so is there a danger the projected uptake there is the most artefactual? Is this what we are seeing in Fig 2 with the model being most out from the data for the lowest decile (projection gives a bigger drop in uptake than happened for the 75+ age group)?

Continuing concern 3: With the caveat that I am not a specialist in these statistical methods: reading appendix B took me many reads to follow what was going on (Figure 7 was very helpful, and I could not have understood the approach from the text alone). I am concerned that it feels somewhat fudge-like, and the sheer number of parameters a, b (one pair for each cohort, and there's 167,424 cohorts). No justification is given for the function U given (though from figure 7, it looks to capture the shape of the blue dots nicely). No justification is given for the method of vertical shift to make the model match the data exactly (as opposed to say, a delta shift or anything else like leaving it as the modelled value, so indeed it doesn't match at $\Delta=0$). The capping of U inside [0,1] I get is needed but would be concerning if used for more than a tiny minority of cohorts: how often was it needed for the chosen D? I slightly wonder by the time there are so many fudges in place, the approach from simulating reduced uptake by tweaking deprivation is really doing that, or if it's just some way to generate a lower uptake from the model which gives reasonable cohort values.

I'm not sure how reassuring figure 2 is: on the one hand, the "predictions" do seem to work well for the different deciles (expect lowest), but is that just a simple reflection of the uptake for 1st booster which would be there for any approach? The results by sex are less encouraging: the first booster rates were very similar for female and male, and so the model predictions are very similar for booster (for 75+), whereas the rates appear to have dropped more for female. So there's maybe a different set of things going on for second booster which hasn't been captured here, and isn't commented on, only the comment given is that the model captures the observed trend, which isn't true for sex.

Smaller observations:

- Fig 1: could do with brief definition of "returning" and "overall" in fig legend (I know it is in main text)
- Section 4.1 starts with "This hypothesis.." and not clear what it is referring to (but I think I can deduce based on rest of paper)
- Section 4.2, language needs to be clear on what is a prediction/projection/extrapolation from the model rather than reported data. E.g. "As nationwide update calls, uptake declines fastest in...". Should be something like "uptake is projected to.." or "our model suggests that.." or similar. This needs a check throughout on what is concrete reporting on data, and what is suggested from this work.

- In discussion statement that model was not informed by Fig 2 needs clarification. Clearly the model was informed by the first booster data (which is on fig 2) and the overall drop in 2nd booster (which does therefore use some of the 2nd booster data). Presumably the intention is to say the model was not informed by the detail of 2nd booster uptakes by SIMD decile or sex.

- In abstract "The high correlation amongst these factors also suggests that, should vaccine uptake decrease, the impact of deprivation is likely to increase.." High correlation of what? Shown where in paper? And why should this high correlation lead to more impact of deprivation? Might be true, but not deducible from what is in the paper as things stand.

- Why is the used value of Delta not given for the fitted results (section 4)?

- First paragraph of paper leads out on vaccine hesitancy being multifactorial, but no citations. Could do with more on what is already known on how the factors considered in this paper influence uptake/hesitancy.

Reviewer #2

(Remarks to the Author)

This manuscript sheds light on the estimated drop in COVID-19 (booster) vaccine uptake by using granular openly available data on socio-demographic status in Scotland in a random forest model. The analysis is presented in a clear and interesting way. Figures are strong and capture the main findings of the authors well. I consider the work scientifically sound and interesting for Communications Medicine readers and the wider scientific community.

The authors have blanked out a part of the data to test the precision of the model and its outputs, which results in good model performance and allows the authors to also use the model for alternative baseline/counterfactual scenarios. In addition to forecasting the drop in booster vaccine uptake, the authors also showcase how publicly available data can be used to produce important estimates that goes also beyond the realm of COVID-19 vaccination.

I have some minor suggestions for the authors to further improve the manuscript:

- The first mention of counterfactual scenarios does not clearly explain what the authors mean. An additional sentence that describes that it is an adjustment of the baseline scenario would guide the readers

- I find the figures in this manuscript extremely informative, but some of the figures could be explained in more detail in the main text. For instance, Figure 3A shows a drop for higher SIMD ranks for all counterfactual scenarios which is not described in the main text. Figure 5 breaks up the deprivation index in single variables to show the importance of the variables/ranks for the prediction of vaccine uptake. This is an extremely important point which is not touched upon in the manuscript at all. I highly encourage the authors to strengthen their description and explanation of figure 5 in the main text and also break down the main explanatory variables when it comes to vaccine uptake. It is clear that age and being male are important, but when it comes to the other variables they are not explained at all, while there are meaningful differences between for instance housing and employment

- In addition it would be helpful for the readers to have a clearer description of the variables/ranks that make up the deprivation index. A table or Annex with an explanation on the main explanatory variables would be helpful.

- Please define the abbreviation BAME in the main text

- It is unclear to me how the explanatory variables age, gender, and the deprivation index are correlated. Could the authors provide additional information on the potential autocorrelation between the variables?

- The authors stress multiple times that there is uncertainty around the population estimates at DZ level and that this impacts the estimates for the overall uptake. It would be helpful if the authors could quantify or give a better sense of how big this uncertainty is and how much it impacts model outcomes.

Thank you very much for the opportunity to review this interesting manuscript,

Reviewer #3

(Remarks to the Author)

This study fits vaccination data in Scotland (specifically booster uptake) using a random forest to learn about the influence of various factors such as age, sex, ethnicity and socio-economic status. The study proceeds to predict the future spatial variation of booster uptake, under different level of overall uptake. I find the study valuable in terms of understanding the risk factors associated with low vaccine uptake. Additionally, the way they conduct the prediction is rather unique and interesting. However, I believe the authors need to (a) be clearer about their work, (b) explain why they use their prediction method and why it works, and (c) to discuss more of the significance of their findings, in terms of potential explanation and in terms of policy.

While the manuscript is generally easy to follow, I find the transition from Section 3 (inference) to Section 4 (forecasting) confusing. The model does not have a time component, thus it is not straightforward for anyone to see how you can use the same model to forecast future rollout. The word choice "to generate counterfactual scenarios" in the start of Section 4 adds to the confusion: counterfactual to what? There wasn't much explanation on it, so the first thing I can think of about the counterfactual scenarios are, for example, the case when people are older, when people are less or more deprived. None of which are suitable for forecasting the future rollout per se.

I think the authors need to start out clarifying their approach vs the "common sense" approach. This is purely based on my guess, but I think the authors intention was that, in a future rollout, if one were to think that the overall vaccine is lower, then

the most straightforward prediction is that there is a uniform decline in uptake. So, if cohort A and B was 80% and 70% uptake, then the future could be a uniform -10% (or perhaps in the logit scale), resulting in 70 and 60% uptake. However, the authors think that the decline wouldn't have been uniform, as a result, they came out with a seemingly "hacky" way, by suggesting that the decline in uptake would follow the gradient along the deprivation ranking. If this is the rationale of using their way of forecasting future rollout, they need to spell it out clearly and early in the manuscript.

I have mentioned that author's method is seemingly "hacky", because I don't think the authors done much in explaining the concept behind their method, and why it would work. It is an unintuitive approach: the authors create future scenarios by adding the deprivation covariate by a delta value. That is like saying that the authors think that in future rollout, the society would have decayed by so much that a rank 1 DZ will become an equivalent of rank 101 DZ (if delta is 100). While it is possible that people become highly unemployed, experience massive loss of access to healthcare with skyrocketing crime, I don't think this is the authors' intention, but the manuscript does not do a good job in explaining that. Instead, I have to make a guess about why shifting deprivation rank would work. It doesn't help that the covariate is a ranking: what happen when the shifted values fall out of range, e.g., < 1 or > 6000 ?

While the authors alluded to the idea that their prediction is doing a good job (in Figure 2) in predicting the variation in Spring 2022 rollout for 75+, they still need to do more to convince the readers that a simplistic alternative wouldn't have done as well. Why should I choose a method that is kind of unintuitive, when I can just assume a uniform decline in natural or logit space in future rollout? It would be helpful for the authors to show that their method is indeed better than assuming uniform decline. Alternatively, the authors should qualify their work by saying that this is an alternative possibility in which more deprived segment of society will experience more decline in booster uptake, along the deprivation gradient.

Finally, I believe that the authors should at least discuss about the "so what" of their finding. So what if we have prediction maps (in Figure 4), are there any surprise that we can learn from them? E.g., new clusters of low uptakes? What would policy makers have done with these maps? Is it any different from knowing the results from Section 3, and that they should focus on persuading younger people in high deprivation zones to take up vaccine in general?

Specific comments

Abstract, last paragraph: "... there is a substantial influence of several deprivation factors and the proportion of BAME residents."

As far as I understand, the study uses a ranking of some form of composite index of deprivation as predictor, not "several deprivation factors" as predictors. Also, this is the first time BAME the acronym is used here, so I think it should be spelt out.

Figure 1: "Values of 'overall' uptake have greater variability..."

The graphic per se does not show the "greater variability" here, and it can mean different thing to different people (higher variance? larger range? bigger cell-to-cell difference?). Perhaps you meant that the trends shown in 'overall' is not as clear cut and consistent as 'returning'.

Section 3, paragraph 3: "... 90.8% of DZ-level variation ..."

How is DZ-level variation calculated from a model that predicts cohort level uptake?

Section 3, paragraph 3

Confusing to jump into Figure 5BCD here when Figure 2 to 4 and 5A weren't even introduced yet?

Section 3: Weirdness in 70-74 age bin

Some censuses do have data suppression practice that causes those older than certain age to be declared as X years old. Would it help to group 70-74 and 75+ together in one bin?

Version 1:

Reviewer comments:

Reviewer #2

(Remarks to the Author)

The authors have addressed all my comments.

Reviewer #3

(Remarks to the Author)

This is a substantial revision of the first draft with additional data and analyses, and I am generally satisfied with the revised manuscript. However, I do have a major comment about the way the stories are presented.

I find that the rationale for the "novel method" to forecast the distribution of second booster uptake lacking. The authors presented the story as if it is natural to apply the novel (and somewhat imperfect) method for forecasting, after the first part of the analysis (which is to explain the distribution of first booster's uptake). As a result, it reads as though the authors just want to present a somewhat arduous way of doing something with the random forest model they created in part 1, and still didn't do a great job in matching the 2024 data. I find myself keep wondering why they think the methods would work as I read through the manuscript.

Coming out of the first analysis, I think the most "natural" way to forecast second booster uptake is to assume that there is a uniform and proportionate decrease. It seems like this natural way of forecasting would have been inaccurate: decrease in second booster uptake is uneven across the age-deprivation landscape. This perhaps led the authors to seek a way to capture the uneven decline that is larger in younger and more deprived demographic. I think prefacing the second analysis with this sort of rationale (and of course showing proof that proportionate decline assumption would be incorrect), will better help the readers to understand the value in the novel method they presented.

Some specific comments:

Fig 2

Since the main message from the figure is used to illustrate the correlation between booster uptake and deprivation, the colour scale for 2B and 2C is simply too large to drive that point (you basically get different shades of blue and yellow and it's hard to distinguish)

Pg 4, Section 3.1: "DZ-level deprivation ranks, by access"

Access to what? Helpful to spell it out at least once in the manuscript (I think there's another occurrence of this in the Intro)

Pg 5: "The model explains 83.1% of between-cohort variation in returning uptake (fit: 85.6%, test: 74.2%), and 90.9% of variation between DZs (calculating uptake for all individuals aggregated by DZ) (fit: 92.7%, test: 83.6%) (Fig S2)."

Good to explain briefly what is the "fit" and "test" here, and why is the main reported % (e.g., the 83.1%) differ from the fit/test values.

Also, I assume you conducted two regressions, one with cohorts as sample, one with DZ as sample (hence the DZ-level variation). The latter wasn't quite clear to me since it's not mentioned in previous paragraph.

Section 4

Brief explanation on why you're comparing two sets of second booster data (2022 and 2024) here.

Fig 3:

It's quite hard to read and follow what exactly you've done here. Would be better if you could create a "workflow" chart that describe the steps stated in Section 4 Paragraph 5.

Pg 7: "The distribution of second boosters represents uptake in the rollout furthest in time from the earlier acute phases of the pandemic."

I don't understand this sentence.

Reviewer #4

(Remarks to the Author)

The authors have done a great job of addressing the comments made by the reviewers. The data updates, modifications to the random forest methodology and additional clarifications provided are highly commendable. I consider the paper suitable for publication.

Reviewer #5

(Remarks to the Author)

In this paper, the authors study long-term spatial patterns in COVID-19 booster vaccine uptake. For the first and second booster programs, they described the variation in uptake across demographics and specific markers for socioeconomic deprivation and showed how that manifested as a spatial clustering of communities with low uptake. They then explored a novel method for using this model for first boosters to predict future spatial variation, under the assumption that the uptake continued to fall and the risk factors for vaccine hesitancy were to persist. The topic of inequalities raised in the paper is very important both in the context of the past pandemic but also in the context of preparedness for similar events in the future or even seasonal vaccination against other pathogens.

I have carefully reviewed the revised version of the manuscript. Major points of the reviewers seem to have been addressed. I still however have some remarks not mentioned before:

- The most significant risk factor for severe COVID-19 outcomes in the post-pandemic era is the presence of specific chronic medical conditions (which is strongly correlated with age). The fact is not mentioned anywhere, as the paper's focus is solely on age and deprivation. In this sense, one could argue that low vaccine uptake in young individuals from deprived neighborhoods is a secondary factor for post-pandemic public health outcomes such as COVID-19 hospitalizations. This is in contrast to the pandemic period when no one or a small proportion of the population had immunity and individuals from deprived neighborhoods indeed suffered a disproportionate burden of severe COVID-19.
- The abstract does not mention that analyses refer to Scotland.
- In the absence of instructions, I could not run the model scripts. Springer Nature has useful checklists for depositing code to ensure their accessibility for a reader.
- If vaccination data are not publicly available, would it be possible to create synthetic data?

Response to reviewers

September 2024

We first acknowledge the significant delay between initial submission, and submission of this revision. The primary reason for this is due to the data we used for the analysis. The data sharing agreement (DSA) was initially made on an emergency basis, however it lapsed prior to receiving reviewer comments. There was a substantial, unexpected delay to negotiating a new (non-emergency) DSA. This has since been granted, however with new data that has required some additional modelling. The delay has however allowed us to look in more detail at the impact of changes in circumstances and its impact on vaccine uptake since 2021, which in our view substantially improves the value of the manuscript. This is especially in the light of a new wave of COVID posing substantial pressures on Scottish healthcare systems as of today, where some of that pressure may be attributable to declining vaccine-induced immunity. We are grateful to the reviewers for their patience.

The initial aim of this work was to predict future patterns in booster vaccine uptake. From elapsed time since initial submission we can now in fact assess those predictions, particularly trends with respect to age and deprivation. In turn we enclose a significantly revised manuscript, updated to reflect the new data, and the fact that we can test our model prediction against the new data. We find the model prediction mainly to under-predict skews with respect to age and deprivation, in a manner we can attribute to the changing context of the epidemic between rounds of vaccination.

We address minor comments in-line below this response. However, Reviewers 1 and 3 in particular expressed concern about how we are generating model predictions. As their concerns are quite broad and overlapping we will provide a single comprehensive response here, rather than going line-by-line.

Why did you choose this approach for predicting uptake? Can you justify why you used this method over other methods? Is it arbitrary? What value does this method bring?

First we agree that we should outline our approach much more clearly, and better justify it, especially given that it is not standard.

The problem we address in this work is how to use existing data on vaccination uptake to predict how longer-term spatial patterns in uptake will look. We are not of course predicting what that uptake will actually be, but instead ask if a given number of boosters are administered, to where/whom will those boosters go to.

We wish to do this prediction at fine-scale. The uneven drops in uptake from the primary course to boosters (i.e. our “returning” uptake) shows a strong sex/age/deprivation trend. Thus we have reason to suspect further drops to be uneven, rather than a uniform drop.

To start, then, a simpler approach such as Reviewer 1’s suggested approach would be to design a “hazard” function for a declining uptake proportion at cohort level P , $P \rightarrow 1 - (1 - P)^x \approx xP - \frac{1}{2}(xP)^2 + \dots$ for some $x \in (0, 1)$. This would of course be a much simpler calculation versus the analysis here, however the hazard function highlights the central problem we faced with simpler approaches. Simply put: *how does one make a non-arbitrary choice for x ?* Further, *how would x vary between different population groups?*

Towards this, our random forest regression model provides two insights:

- (i) a remarkable amount of variation in booster uptake across population groups can be explained by variation in population structure (age, sex, BAME residents) and community deprivation;
- (ii) for some cohorts, the fit value of booster uptake was very sensitive to small changes in the community deprivation, whereas for other cohorts the fit value was more robust to changes.

Our hypothesis, then, is that deprivation is the key driver for *spatial* differences in vaccine uptake (not age, sex or ethnicity in the same manner, as each have much more limited spatial heterogeneity), and that those cohorts whose fit uptake are more sensitive to small changes in community deprivation

are prone to suffer disproportionately high falls in uptake in the future (analagous to smaller values of “ x ” from above). Thus our choice of x in this manner is a scientifically motivated one.

With random forest models, a standard method for assessing the impact of different variables on the model prediction is looking at the *partial dependencies* (a similar approach being *accumulated local effects*) which assess the average model prediction when a given variable assumes a certain value?

Our method for using our model to predict lower-uptake distributions is certainly novel, but is effectively an extension of this method. We have modified it to:

- (i) manage with changing many variables instead of just one; we have seven metrics for deprivation, all of which are important, and;
- (ii) explore values that exceed the training set; random forest models perform poorly with values that exceed what was in the training set (e.g. a rank of “-386”), which makes it necessary here fit a curve to the known range, to extrapolate to lower uptake scenarios.

To point (ii) we fit a sigmoid function; a two parameter function that is continuous, and is bound between 0 and 1. As we wish to explore fine-scale variation, it is necessary to have a fit for each cohort. On reflection we think that the analysis would have benefited from our predictions having associated sensitivity analyses/uncertainty bounds, especially for individual-cohort prediction.

Finally, on Reviewer 3’s concerns as to what value can be drawn from such an approach. Simply put, for individual-based models, the profile of vaccine-induced immunity is one of few variables that modellers could know precisely based on data that are routinely gathered. The method we discuss here is a way of precisely modelling longer-term vaccination uptake that can feed into such disease models, based on current data. Data on the different stages of disease progression (with early COVID-19 the exception) are otherwise generally sparse and low resolution by virtue of them being difficult to gather (taking influenza as an example, where incidence is measured by e.g. the volume of phonecalls to the NHS citing flu-like symptoms). Vaccination data, on the other hand, are routinely collected for many diseases, but very rarely made available to researchers.

Summary of changes

Due to the size of the revision, we have not marked up individual changes in the document. To summarise the overall changes we have made to the manuscript compared to the previous version:

- We have rewritten Section 4 in its entirety to better describe our method, and justify why we have used it, in the context of extending the idea of partial dependencies
- We have moved the figure for “cohort-level predictions” into the main text to make clearer what we are doing, without the reader having to go to the supplementary material.
- We have removed the predictions for 70%, 60%, 50% uptake, etc., to instead focus on predicting the observed trend in second boosters (which we did not have before), and discussing why the model failed to reproduce it.
- We have rewritten the Discussion section to first explain patterns in the data (i.e., what actually happened), why our model failed to reproduce the trend in second boosters, and how such a method could be applied in the context of infectious disease modelling more broadly.

Finally please find below a set of in-line responses to minor comments.

Anthony Wood (on behalf of all authors)

Changes in data sources

Below is a summary of the changes in data sources:

Vaccination

- **Original:** Dated August 2022. A list of individual vaccinations, where individuals are given a pseudonymised ID with associated age range (5-year ranges), sex, and residing datazone. The pseudonymised ID allowed us to track repeat doses.
- **Revised:** Dated May 2024 For each cohort of individuals grouped by age range (10-year ranges), sex, residing datazone, the number of individuals to have received exactly 1 dose, exactly 2 doses, exactly 3 doses, and exactly 4 doses. Groups with fewer than 5 individuals are labelled '< 5'.

Population

- **Original:** 2021 Small Area Population Estimates:
<https://www.nrscotland.gov.uk/statistics-and-data/statistics/statistics-by-theme/population/population-estimates/small-area-population-estimates-2011-data-zone-based/mid-2021>
- **Revised:** 2022 Census, table UV102b – Population by Age and Sex:
<https://www.scotlandscensus.gov.uk/documents/2022-output-area-data/>

Ethnicity

- **Original:** 2011 Census, table LC2101SC – Ethnic group by Age
<https://www.scotlandscensus.gov.uk/documents/2011-census-table-data-sns-data-zone-2011/>
- **Revised:** 2022 Census, table UV201b – Ethnic group by Age:
<https://statistics.ukdataservice.ac.uk/dataset/scotland-s-census-2022-uv201b-ethnic-group-19->

Reviewer 1

This paper presents an analysis of a very finely scaled data set for COVID-19 vaccination in Scotland. Specifically, it considers first and second booster uptake stratified by fine spatial scale, age, gender and socio-economic factors. There are three main claims:

1. 1st Booster uptake predicts 2nd booster uptake (though at lower general level)
2. Location factors are important for explaining variability in uptake, specifically measures of deprivation and also BAME proportions
3. If uptake decreases, socio-economic factors become relatively more important

The first result is unsurprising but reassuring. The second result is also unsurprising, but very important and there are few studies with the depth of data to explore this well. The third result implies the that the disproportional effects of COVID-19 would be exacerbated, which is clearly important.

The authors are careful not to claim application beyond the specific circumstance studied, but in practice what is learnt from this data will potentially be informative for much wider application. I set out some queries and concerns below. Of all the points below, point 3 is the one that I have the most concerns about, though some aspects of it certainly could be my misunderstanding.

Despite these concerns, this is potentially an important paper as it gives an analysis of an exquisite dataset, and any insights (even with many caveats) about vaccine uptake variation across the population remains among the important things that we should learn from COVID-19, with potential consequences for vaccine control for further COVID-19 variants and other diseases in future. The ability to explore in superb depth here is unique as far as I have seen so far (see for example the work on the residuals in fig 6 by space, as well as the detailed consideration of different deprivation ranks). The approach taken overall could indeed be of use in adapting to other datasets, as the authors mention in the abstract.

1. **The population denominators:** The caution that the 2011 census data is used is noted, and is surely a reasonable approach. Is it possible though to give any early estimates about how far these differ from reality in 2021?

Looking at mid-year estimates on National Records of Scotland (e.g. via <https://scotland.shinyapps.io/nrs-population-estimates/>) it looks like the older population age groups are growing, consistent with an aging population.

The figures show a clear anomaly for the age band 70-74 (e.g Fig 1 second panel), suggesting that uptake appears low as the denominator is erroneously large. But why this one band is so different, and why the wrong way around from the general picture above of an aging population (so the 2011 denominators should be too small at this age group)? Is there any other reason why that age band is such an obvious anomaly? *Authors: Moving to the more accurate 2022 census population (which does not rely on estimating from the 2011 census) has cleared this visual anomaly. It is also not clear to us why this was anomaly was there in the first place (suspecting an undercount of 70-74 and overcount of 75+), aside from the estimates gradually drifting from the “true” value over successive years since the 2011 census. This drift was most apparent in 70-74 as “true” uptake was high enough that doses exceeded the population error (say, there may have been an undercount in the 30-34 population but not revealed as “true” uptake was also far below 100%).*

2. **Interpretation of results for BAME:** Last paragraph of section 3 highlights variability in Scotland of BAME proportions across DZs, and that previous studies have highlighted lower vaccine uptake in BAME communities. The results given verbally in that paragraph confirm that for the fit here, proportion BAME in a DZ is selected by the model. But, the strength of the effect, or even the direction of the effect is not given.

While I know this can't just be listed as a single number out of a random forest approach, every other factor involved is illustrated somewhere – is it possible to give something, e.g. uptake by proportion BAME (possibly binned), given as data and model?

As things stand, the paper justifies the claim that proportion BAME is a factor that influences uptake, but nothing at all on the direction of the effect. Seems important, given that the overarching motivation of identifying how vaccine hesitancy varies across the population. *Authors: To address this we have added to the importance analysis a *partial dependency* analysis, which effectively adds a “directionality” to the importance of each variable. This shows that, while (as the reviewer notes) we can not disentangle a single variable from an approach like this, the effect is that increasing proportions of BAME residents is indicative of lower uptake.*

3. The is my main (and only) major concern: **how the model is used to extrapolate to generate a counterfactual**. Here is my understanding: as this is a statistical random forest model, there is no direct way to tweak a mechanistic parameter to simulate a general broad reduction in vaccine uptake. So, the authors have taken an approach of supposing that everyone is moved to a lower deprivation rank (with a fix to stop this going less than 1), and this is used to create a lower uptake. My concerns are two-fold.

(a) Firstly that this seems odd decision on approach, on the face of it, and not clearly justified.

- This whole approach to simulating a reduction in uptake via turning up deprivation doesn't seem to have been justified. It does achieve the desired outcome of a lower uptake output from model. But why via deprivation? Alternative (equally unjustified) would be to do the same with age (pretend everyone is X years younger, with a fix to keep them over age 20). (One of many problems with that is that it would be extrapolating from semi-fit for the 75+ 2nd boosters to other age groups.) But also why not an even simpler approach: suppose a person in a given cohort has probability P of getting first booster, then if we reduce a pressure/hazard of getting vaccinated, then we can scale new P as $1 - (1 - P)^x$ for some scaling parameter x . This does bake in a model of how vaccine hesitancy will scale between different people. I'm not sure I'm suggesting exactly this model, but how is this less justified than assuming hesitancy scales like everyone shifts deprivation factor in the given way?

(b) Secondly, I wonder if there's a danger of circularity or bias on the result that deprivation becomes more important if uptake drops.

- One of the key points of this paper is that as uptake decreases, socio-economic factors become more important, and vaccine uptake even more disproportionate between groups. However, given the way reduced uptake is simulated is to project from tweaking deprivation rank, this result appears compromised by the approach being so entangled with the claimed effect. In particular, the handling of the most deprived groups requires a fix, so is there a danger the projected update there is the most artefactual? Is this what we are seeing in Fig 2 with the model being most out from the data for the lowest decile (projection gives a bigger drop in uptake than happened for the 75+ age group)?

Continuing concern 3: With the caveat that I am not a specialist in these statistical methods: reading appendix B took me many reads to follow what was going on (Figure 7 was very helpful, and I could not have understood the approach from the text alone). I am concerned that it feels somewhat fudge-like, and the sheer number of parameters a , b (one pair for each cohort, and there's 167,424 cohorts). No justification is given for the function U given (though from figure 7, it looks to capture the shape of the blue dots nicely). No justification is given for the method of vertical shift to make the model match the data exactly (as opposed to say, a delta shift or anything else like leaving it as the modelled value, so indeed it doesn't match at $\Delta = 0$). The capping of U inside $[0,1]$ I get is needed but would be concerning if used for more than a tiny minority of cohorts: how often was it needed for the chosen D ? I slightly wonder by the time there are so many fudges in place, the approach from simulating reduced

update by tweaking deprivation is really doing that, or if it's just some way to generate a lower uptake from the model which gives reasonable cohort values. I'm not sure how reassuring figure 2 is: on the one hand, the "predictions" do seem to work well for the different deciles (expect lowest), but is that just a simple reflection of the uptake for 1st booster which would be there for any approach? The results by sex are less encouraging: the first booster rates were very similar for female and male, and so the model predictions are very similar for booster (for 75+), whereas the rates appear to have dropped more for female. So there's maybe a different set of things going on for second booster which hasn't been captured here, and isn't commented on, only the comment given is that the model captures the observed trend, which isn't true for sex.

Smaller observations:

- Fig 1: could do with brief definition of "returning" and "overall" in fig legend (I know it is in main text) *Authors: Added to figure caption.*
- Section 4.1 starts with "This hypothesis.." and not clear what it is referring to (but I think I can deduce based on rest of paper) *Authors: Edited out in revised version.*
- Section 4.2, language needs to be clear on what is a prediction/projection/extrapolation from the model rather than reported data. E.g. "As nationwide uptake falls, uptake declines fastest in...". Should be something like "uptake is projected to.." or "our model suggests that.." or similar. This needs a check throughout on what is concrete reporting on data, and what is suggested from this work. *Authors: Addressed in revised version.*
- In discussion statement that model was not informed by Fig 2 needs clarification. Clearly the model was informed by the first booster data (which is on fig 2) and the overall drop in 2nd booster (which does therefore use some of the 2nd booster data). Presumably the intention is to say the model was not informed by the detail of 2nd booster uptakes by SIMD decile or sex. *Authors: The model is trained on first boosters only, to predict the distribution of second/future boosters. Here we compare our prediction to those data. As the previous point, this should be more clear in our revised version.*
- In abstract "The high correlation amongst these factors also suggests that, should vaccine uptake decrease, the impact of deprivation is likely to increase.." High correlation of what? Shown where in paper? And why should this high correlation lead to more impact of deprivation? Might be true, but not deducible from what is in the paper as things stand. *Authors: We do now include a correlation plot in the Appendix. However in any case, in the revision this line has been removed.*
- Why is the used value of Delta not given for the fitted results (section 4)? *Authors: We have now included this.*
- First paragraph of paper leads out on vaccine hesitancy being multifactorial, but no citations. Could do with more on what is already known on how the factors considered in this paper influence uptake/hesitancy. *Authors: We have added further citations and discussion to the Introduction section.*

Reviewer 2

This manuscript sheds light on the estimated drop in COVID-19 (booster) vaccine uptake by using granular openly available data on socio-demographic status in Scotland in a random forest model. The analysis is presented in a clear and interesting way. Figures are strong and capture the main findings of the authors well. I consider the work scientifically sound and interesting for Communications Medicine readers and the wider scientific community. The authors have blanked out a part of the data to test the precision of the model and its outputs, which results in good model performance and allows the authors to also use the model for alternative baseline/counterfactual scenarios. In addition to forecasting the drop in booster vaccine uptake, the authors also showcase how publicly available data can be used to produce important estimates that goes also beyond the realm of COVID-19 vaccination. I have some minor suggestions for the authors to further improve the manuscript:

- The first mention of counterfactual scenarios does not clearly explain what the authors mean. An additional sentence that describes that it is an adjustment of the baseline scenario would guide the readers *Authors: We have made this more clear in the wider revision of the section on uptake prediction (see initial response for details).*
- I find the figures in this manuscript extremely informative, but some of the figures could be explained in more detail in the main text. For instance, Figure 3A shows a drop for higher SIMD ranks for all counterfactual scenarios which is not described in the main text. Figure 5 breaks up the deprivation index in single variables to show the importance of the variables/ranks for the prediction of vaccine uptake. This is an extremely important point which is not touched upon in the manuscript at all. I highly encourage the authors to strengthen their description and explanation of figure 5 in the main text and also break down the main explanatory variables when it comes to vaccine uptake. It is clear that age and being male are important, but when it comes to the other variables they are not explained at all, while there are meaningful differences between for instance housing and employment *Authors: We have expanded discussion of this in the context of partial dependency plots, showing clearly for all variables the effects stratified by age.*
- In addition it would be helpful for the readers to have a clearer description of the variables/ranks that make up the deprivation index. A table or Annex with an explanation on the main explanatory variables would be helpful. *Authors: We have added detail in the Appendix.*
- Please define the abbreviation BAME in the main text *Authors: We have now done this.*
- It is unclear to me how the explanatory variables age, gender, and the deprivation index are correlated. Could the authors provide additional information on the potential autocorrelation between the variables? *Authors: These are indeed highly correlated, which motivated our use of random forests. We have included a new correlation plot in the Appendix.*
- The authors stress multiple times that there is uncertainty around the population estimates at DZ level and that this impacts the estimates for the overall uptake. It would be helpful if the authors could quantify or give a better sense of how big this uncertainty is and how much it impacts model outcomes. *Authors: We now use 2022 census population data rather than an estimate, which removes the anomaly in Figure 1. Model outcomes are unaffected as we fit to returning uptake, where the denominator is the number of people to have received any dose, rather than a population.*

Thank you very much for the opportunity to review this interesting manuscript.

Reviewer 3

This study fits vaccination data in Scotland (specifically booster uptake) using a random forest to learn about the influence of various factors such as age, sex, ethnicity and socio-economic status. The study proceeds to predict the future spatial variation of booster uptake, under different level of overall uptake. I find the study valuable in terms of understanding the risk factors associated with low vaccine uptake. Additionally, the way they conduct the prediction is rather unique and interesting. However, I believe the authors need to

1. be clearer about their work
2. explain why they use their prediction method and why it works
3. to discuss more of the significance of their findings, in terms of potential explanation and in terms of policy.

While the manuscript is generally easy to follow, I find the transition from Section 3 (inference) to Section 4 (forecasting) confusing. The model does not have a time component, thus it is not straightforward for anyone to see how you can use the same model to forecast future rollout. The word choice "to generate counterfactual scenarios" in the start of Section 4 adds to the confusion: counterfactual to what? There wasn't much explanation on it, so the first thing I can think of about the counterfactual scenarios are, for example, the case when people are older, when people are less or more deprived. None of which are suitable for forecasting the future rollout per se.

I think the authors need to start out clarifying their approach vs the "common sense" approach. This is purely based on my guess, but I think the authors intention was that, in a future rollout, if one were to think that the overall vaccine is lower, then the most straightforward prediction is that there is a uniform decline in uptake. So, if cohort A and B was 80% and 70% uptake, then the future could be a uniform -10% (or perhaps in the logit scale), resulting in 70 and 60% uptake. However, the authors think that the decline wouldn't have been uniform, as a result, they came out with a seemingly "hacky" way, by suggesting that the decline in uptake would follow the gradient along the deprivation ranking. If this is the rationale of using their way of forecasting future rollout, they need to spell it out clearly and early in the manuscript.

I have mentioned that author's method is seemingly "hacky", because I don't think the authors done much in explaining the concept behind their method, and why it would work. It is an unintuitive approach: the authors create future scenarios by adding the deprivation covariate by a delta value. That is like saying that the authors think that in future rollout, the society would have decayed by so much that a rank 1 DZ will become an equivalent of rank 101 DZ (if delta is 100). While it is possible that people become highly unemployed, experience massive loss of access to healthcare with skyrocketing crime, I don't think this is the authors' intention, but the manuscript does not do a good job in explaining that. Instead, I have to make a guess about why shying deprivation rank would work. It doesn't help that the covariate is a ranking: what happen when the shiyed values fall out of range, e.g., < 1 or > 6000 ?

While the authors alluded to the idea that their prediction is doing a good job (in Figure 2) in predicting the variation in Spring 2022 rollout for 75+, they still need to do more to convince the readers that a simplistic alternative wouldn't have done as well. Why should I choose a method that is kind of unintuitive, when I can just assume a uniform decline in natural or logit space in future rollout? It would be helpful for the authors to show that their method is indeed better than assuming uniform decline. Alternatively, the authors should qualify their work by saying that this is an alternative possibility in which more deprived segment of society will experience more decline in booster uptake, along the deprivation gradient.

Finally, I believe that the authors should at least discuss about the "so what" of their finding. So what if we have prediction maps (in Figure 4), are there any surprise that we can learn from them? E.g., new clusters of low uptakes? What would policy makers have done with these maps? Is it any different from knowing the results from Section 3, and that they should focus on persuading younger people in high deprivation zones to take up vaccine in general?

Specific comments

- Abstract, last paragraph: "... there is a substantial influence of several deprivation factors and the proportion of BAME residents." As far as I understand, the study uses a ranking of some form of composite index of deprivation as predictor, not "several deprivation factors" as predictors. Also, this is the first time BAME the acronym is used here, so I think it should be spelt out. *Authors: We now write BAME out in full. Overall deprivation (i.e. the decile) is a composite measure of multiple deprivation, but the model is trained with ranks across more granular aspects of deprivation (access, education etc.) We have revised the abstract to refer to "community deprivation".*
- Figure 1: "Values of 'overall' uptake have greater variability..." The graphic per se does not show the "greater variability" here, and it can mean different thing to different people (higher variance? larger range? bigger cell-to-cell difference?). Perhaps you meant that the trends shown in 'overall' is not as clear cut and consistent as 'returning'. *Authors: This has been rectified with the change in data source, however it was clumsy wording on our behalf; we meant variability in the sense if noise introduced from the population denominators being likely inaccurate (and increasingly so as the cohort becomes smaller).*
- Section 3, paragraph 3: "... 90.8% of DZ-level variation ..." How is DZ-level variation calculated from a model that predicts cohort level uptake? *Authors: This is from aggregating predictions for all cohorts to calculate uptake of the DZ as a whole. We have clarified this.*
- Section 3, paragraph 3 Confusing to jump into Figure 5BCD here when Figure 2 to 4 and 5A weren't even introduced yet? Section 3: *Authors: Figures are now referred in order.*
- Weirdness in 70-74 age bin Some censuses do have data suppression practice that causes those older than certain age to be declared as X years old. Would it help to group 70-74 and 75+ together in one bin? *Authors: This is no longer an issue with the new population data.*

Feedback from reviewers and responses

April 2025

Reviewer 2

The authors have addressed all my comments.

Reviewer 3

This is a substantial revision of the first draft with additional data and analyses, and I am generally satisfied with the revised manuscript. However, I do have a major comment about the way the stories are presented.

I find that the rationale for the “novel method” to forecast the distribution of second booster uptake lacking. The authors presented the story as if it is natural to apply the novel (and somewhat imperfect) method for forecasting, after the first part of the analysis (which is to explain the distribution of first booster’s uptake). As a result, it reads as though the authors just want to present a somewhat arduous way of doing something with the random forest model they created in part 1, and still didn’t do a great job in matching the 2024 data. I find myself keep wondering why they think the methods would work as I read through the manuscript.

Coming out of the first analysis, I think the most “natural” way to forecast second booster uptake is to assume that there is a uniform and proportionate decrease. It seems like this natural way of forecasting would have been inaccurate: decrease in second booster uptake is uneven across the age-deprivation landscape. This perhaps led the authors to seek a way to capture the uneven decline that is larger in younger and more deprived demographic. I think prefacing the second analysis with this sort of rationale (and of course showing proof that proportionate decline assumption would be incorrect), will better help the readers to understand the value in the novel method they presented.

Authors: This is broadly our idea, yes. Briefly: a proportional decrease is not a reasonable assumption, and our method provides a way of modelling a (non-arbitrary) disproportionate decrease, based on trends in uptake data at the time (initial course/first boosters only, the data we have fit the model to) and our hypothesis that spatial differences in uptake are driven by deprivation.

More comprehensively, suppose a modeller is generating outbreak scenarios, where vaccine uptake is lower than it is today. For a disease like COVID-19 the risk of severe disease differs substantially from person to person, and in turn model outputs such as hospital occupancy are highly sensitive to vaccination uptake in those most vulnerable. Therefore it is important to correctly calibrate uptake rates in those specific vulnerable populations.

The easiest way of modelling uptake here would of course be to take uptake as it is today, decrease it proportionately, and input it into the model. However the demographic skew in people failing to return for a first booster (Fig 1a, Returning) makes clear that, as the reviewer says, this is not a reasonable assumption and may lead to inaccurate model outcomes.

The next step then is decrease uptake in a way where the fall is disproportionate across different demographics. However, without a wealth of longitudinal data (at the time the model was fit), to avoid an arbitrary choice for this disproportionality we instead proposed and applied the method described in the manuscript. We acknowledge that it is quite elaborate but we hope in this revision and the previous we have made it clearer, and also explained why we think it performed poorly on the 2024 data; the model is based on uptake near the height of the pandemic, and public attitudes/incentives/motivation towards choosing to be vaccinated were very different to today.

We have added a paragraph to the beginning of Section 4 *A method for predicting future distributions of uptake* to emphasise why we are using our method, over a simpler method.

Some specific comments:

- Fig 2; Since the main message from the figure is used to illustrate the correlation between booster uptake and deprivation, the colour scale for 2B and 2C is simply too large to drive

that point (you basically get different shades of blue and yellow and it's hard to distinguish)

Authors: We have changed the colour scale from 0–100% to 50–100% to improve the contrast.

- Pg 4, Section 3.1: "DZ-level deprivation ranks, by access" Access to what? Helpful to spell it out at least once in the manuscript (I think there's another occurrence of this in the Intro)

Authors: We have elaborated on access in the main text (we believe the others are mostly self-explanatory), and referred the reader to the supplementary information where a more detailed description of all deprivation measures is provided.

- Pg 5: "*The model explains 83.1% of between-cohort variation in returning uptake (fit: 85.6%, test: 74.2%), and 90.9% of variation between DZs (calculating uptake for all individuals aggregated by DZ) (fit: 92.7%, test: 83.6%) (Fig S2).*" Good to explain briefly what is the "fit" and "test" here, and why is the main reported % (e.g., the 83.1%) differ from the fit/test values. Also, I assume you conducted two regressions, one with cohorts as sample, one with DZ as sample (hence the DZ-level variation). The latter wasn't quite clear to me since it's not mentioned in previous paragraph.

Authors: Yes this is correct, from our single model (which is fit to cohorts, the smallest subunit), here we have looked at variance explained at the cohort level, and also the coarser datazone level (which is more representative of spatial variation explained). We have reworded this paragraph in the main text to make this clearer.

- Section 4; Brief explanation on why you're comparing two sets of second booster data (2022 and 2024) here.

Authors: We have added a paragraph to explain this.

- Fig 3: It's quite hard to read and follow what exactly you've done here. Would be better if you could create a "workflow" chart that describe the steps stated in Section 4 Paragraph 5.

Authors: We have added a workflow.

- Pg 7: "The distribution of second boosters represents uptake in the rollout furthest in time from the earlier acute phases of the pandemic." I don't understand this sentence.

Authors: We agree this is unclear and have reworded it – we are making the point that second boosters, being the most recent round of data we have, are likely closest to what "typical" uptake will be for boosters in the long term as the extraordinary pressures from the pandemic have subsided.

Reviewer 4

The authors have done a great job of addressing the comments made by the reviewers. The data updates, modifications to the random forest methodology and additional clarifications provided are highly commendable. I consider the paper suitable for publication.

Reviewer 5

In this paper, the authors study long-term spatial patterns in COVID-19 booster vaccine uptake. For the first and second booster programs, they described the variation in uptake across demographics and specific markers for socioeconomic deprivation and showed how that manifested as a spatial clustering of communities with low uptake. They then explored a novel method for using this model for first boosters to predict future spatial variation, under the assumption that the uptake continued to fall and the risk factors for vaccine hesitancy were to persist. The topic of inequalities raised in the paper is very important both in the context of the past pandemic but also in the context of preparedness for similar events in the future or even seasonal vaccination against other pathogens.

I have carefully reviewed the revised version of the manuscript. Major points of the reviewers seem to have been addressed. I still however have some remarks not mentioned before:

- The most significant risk factor for severe COVID-19 outcomes in the post-pandemic era is the presence of specific chronic medical conditions (which is strongly correlated with age). The fact is not mentioned anywhere, as the paper’s focus is solely on age and deprivation. In this sense, one could argue that low vaccine uptake in young individuals from deprived neighborhoods is a secondary factor for post-pandemic public health outcomes such as COVID-19 hospitalizations. This is in contrast to the pandemic period when no one or a small proportion of the population had immunity and individuals from deprived neighborhoods indeed suffered a disproportionate burden of severe COVID-19.

Authors: With the data available to us we stratify the population by community deprivation and age, though of course these are not “causative” factors for severe COVID-19. We agree then that pre-existing health conditions are the main causative risk factor for severe disease now that population immunity has built up. While age is dominant there is also significant correlation with deprivation (as evidenced by metrics in the SIMD e.g. the combined measure for health deprivation including the proportion of individuals on incapacity or disability benefit).

We have revised the text in the introduction to emphasise that the underlying causative risk factor are pre-existing health conditions, with age and deprivation strongly correlated.

- The abstract does not mention that analyses refer to Scotland.

Authors: Scotland is now referred to in the abstract.

- In the absence of instructions, I could not run the model scripts. Springer Nature has useful checklists for depositing code to ensure their accessibility for a reader.

Authors: We have revised the model script to run closed-form with a synthetic dataset, and have checked it runs on an independent machine.

- If vaccination data are not publicly available, would it be possible to create synthetic data?

Authors: As above, we have provided a synthetic vaccination dataset with the broad deprivation-level patterns captured, to run with the above code. All other data used are public.